# Application Research on Risk Assessment of Municipal Pipeline Network Based on Random Forest Machine Learning Algorithm

Hang Cen [1], Delong Huang [1,*], Qiang Liu [2], Zhongling Zong [1] and Aiping Tang [2]

1  School of Civil and Ocean Engineering, Jiangsu Ocean University, Lianyungang 222005, China;
   cen1364570502@gmail.com (H.C.); jouzongzhl@jou.edu.cn (Z.Z.)
2  School of Civil Engineering, Harbin Institute of Technology, Harbin 150090, China; qiangliu_hit@163.com (Q.L.);
   tangap@hit.edu.cn (A.T.)
*  Correspondence: huang06080601@163.com; Tel.: +86-151-5094-1271

**Abstract:** Urban municipal water supply is an important part of underground pipelines, and their scale continues to expand. Due to the continuous improvement in the quality and quantity of data available for pipeline systems in recent years, traditional pipeline network risk assessment cannot cope with the improvement of various monitoring methods. Therefore, this paper proposes a machine learning-based risk assessment method for municipal pipe network operation and maintenance and builds a model example based on the data of a pipeline network base in a park in Suzhou. We optimized the random forest learning model, compared it with other centralized learning methods, and finally evaluated the model's learning effect. Finally, the risk probability associated with each pipe segment sample was obtained, the risk factors affecting the pipe segment's failure were determined, and their relevance and importance ranking was established. The results showed that the most influential factors are pipe material, soil properties, service life, and the number of past failures. The random forest algorithm demonstrated better prediction accuracy and robustness on the dataset.

**Keywords:** machine learning; random forest; municipal pipeline network; monitoring data; risk assessment

## 1. Introduction

As the urban population grows, new pipes are added to the existing network, which results in a heterogeneous system with components of different ages, materials, and sizes, making operation and maintenance monitoring an arduous task. The leak detection technology of the water supply network has been developed in the past two decades. Currently, the commercial sector offers a range of hardware-based leak detection devices [1]. In recent years, new research on risk assessment strategies based on algorithms or models has been proposed. Whether under steady-state or transient conditions [2,3], each of the different techniques proposed so far has varying degrees of performance efficiency, depending on the external or internal system conditions and the limitations of the method itself.

The earliest model-based technique was proposed by Pudar as early as 1992, where leak detection was formulated as a least squares parameter estimation problem [4]. Using a multivariate Gaussian mixture-based graphical model and geostatistical techniques, Romano et al., utilized data from more piezometers to identify potential locations of pipe segment ruptures within the metered area [5]. Nevertheless, since pressure is not sensitive to rupture [6] and often stochastic demand fluctuations can mask hydraulic changes induced by blasting, these methods can only narrow down the potential location of damage to a rough area of hundreds of meters consisting of at least a few dozen pipes, insufficient to timely repair squib and resume use [7]. To more accurately locate pipeline failures, new, targeted monitoring is often required to collect detailed information (e.g., acoustic or transient signals [8–12]) to detect abnormal system behavior at potential locations [13]. Acoustic devices such as earpieces and noise correlators are widely used for human inspection. Although these measures are effective and easy to implement, they are labor-intensive

methods. This method proposed by Kang et al., can automatically detect burst/leak locations by temporarily placing acceleration sensors and analyzing acoustic signals [11]. Xuan Hu et al., proposed a novel leak detection model (DBSCAN-MFCN) based on density-spatial clustering applied to noise (DBSCAN) and a multi-scale fully convolutional network (MFCN) to manage water loss [14]. However, its reliability is sensitive to the characteristics of the leak condition (e.g., water pressure, leak flow), and the detection range is limited by the clarity and correlation of the acoustic signal. Transient-based methods locate bursts by analyzing characteristics of transient waves such as propagation, reflection, and damping [15]. However, background noise or other activities in the system may affect the burst-induced transient signal, especially when the number of pipelines to be analyzed increases [16]. Furthermore, most of these methods require computational numerical simulations and use high-frequency sensors, which are costly. Therefore, existing acoustic or transient-based methods are not suitable for accurate pulse localization from a rather large area (such as DMA or dozens of pipelines) to one pipeline.

With the further development of online monitoring equipment that can collect flow and pressure data in near real-time [17–19]. Creaco et al., mentioned in the article that the real-time monitoring system has advantages in water pressure control [19]. Data-driven risk assessment methods are less costly and more labor and time efficient [20]. Once calibrated, optimized, and validated, the pipeline system is continuously monitored using the developed risk assessment algorithm without many operators, mainly by the computer system, making the maintenance of the network model very efficient.

At present, the risk assessment of municipal pipeline networks mainly focuses on pipeline risk assessment, qualitative and quantitative analysis of accident consequences [21], and related research and applications are mainly carried out by developed countries such as Europe and the United States.

Machine learning, which uses big data to extract information, as a predictive system, has received more and more attention in this field [22]. Among probability-based approaches, Bayesian Belief Networks (BBNs) have received particular attention in the scientific literature [23]. BBNs are based on the Bayesian probability theorem. Its structure is represented by a directed acyclic graph, where nodes represent parameters and arcs represent probabilistic relationships between parameters. They allow for predictive and diagnostic reasoning [24]. Its main disadvantage is that the interpretation of the results is not straightforward and often requires expert opinion to generate conditional probabilities.

Since Mounce et al., introduced artificial neural networks (ANNs) into the detection of emergencies [25], the method of machine learning applied to pipeline risk prediction and assessment has developed rapidly. A rule-based system was used to evaluate the deviation between the observed data and the data modeled by ANNs for detecting emergencies [26], followed by fuzzy inference methods combined with artificial neural networks for classification [27].

The artificial neural network approach attempts to mimic the functioning of the human brain. Neurons are represented by nodes, and nerve impulses are represented by the weighted sum of each neuron's input values. The learning of the network is achieved by adjusting these weights, while the structure of the network usually does not change [28]. Although the artificial neural network has good pattern recognition and generalization ability, it cannot explain the relationship between parameters [29]. According to the research conclusion of Shridhar Yamijala, the logistic regression (LR) algorithm is more suitable for solving this problem, and it can provide a reliable basis for estimating the rupture probability of a given pipe segment [30].

Chen et al., established a model evaluation model based on the random forest model, verified the feasibility of the model through a rainstorm prediction example in Nanjing, China, and finally concluded that the risk of rainstorm disasters is regional [31]. Zahedi et al., proposed the use of a machine-learning algorithm based on random forests to predict the erosion of solid particles on elbows [2]. The correlation between different influencing factors and erosion volume was obtained. For example, the gas velocity, particle size,

and hardness have an effective positive correlation with the erosion amount, while the liquid viscosity, liquid flow rate, and pipe diameter have a negative correlation with the erosion amount. However, no evidence has been found of its application to pipeline feature classification to predict future failures. This approach was proposed in this study because it has proven to be suitable for handling imbalanced data [32] and has achieved successful results in other research areas [33]. The investigation of risk factors in urban utility networks encounters two primary challenges: First, pipe network facilities are interdependent on multiple levels, but they are often constructed and maintained by different companies or departments, and these stakeholders independently plan and implement street works without regard to interdependencies. A problem with one construction project can damage other nearby facilities, creating cascading problems [34]. Second, practical applications suffer from a scarcity of efficient samples and an excessive reliance on expert knowledge, resulting in inherent constraints during the development of specific urban utility network maintenance risk identification methodologies. Addressing these challenges will pave the way for a more proactive approach to urban infrastructure maintenance.

To solve the above problems, this paper proposes a risk assessment method for pipeline network operation and maintenance based on machine learning, which is used to solve the risk management of urban pipeline networks based on intelligent monitoring and data-driven, as well as the assessment of risk factors related to the underground pipeline network.

This method adopts the inference method of machine learning, which can obtain comprehensive evaluation results when more artificial decisions or real data are input. Quantify risk factors through machine learning, the risk factors and potential consequences of pipeline failure states are inferred based on historical data and data retrieved from an integrated database. An example of a risk assessment model for municipal pipeline network operation and maintenance is established. The model is established based on the data of the pipeline network in Suzhou Industrial Park. The learning model of the random forest method of the integrated learner is constructed, and the pipeline attribute-pipeline normal/accident type data is used for training, and the pipeline attribute conditions are input to obtain the classification of each pipeline condition in the prediction result; by evaluating the performance discrimination of the model, the good performance of the random forest algorithm is determined, and further risk assessment and analysis are carried out based on the model results.

## 2. Risk Factors of Pipeline

In most cases, it is impossible to obtain all the data that affect the operation and state of the network. However, over the past few years, the availability of data for this industry has increased due to the development of new technologies and growing interest in the usefulness of big data. This enables an in-depth study of variables affecting pipeline failures and leads to the application of different predictive models. The introduction of geographic information system (GIS) tools provides a new perspective on the storage, manipulation, and access of water network data. In fact, many research case studies are based on data extracted from such tools.

Every water supply network has different physical, operational, and environmental characteristics. Physical characteristics are characteristics of the network layout and state, such as the material, diameter, or age of the pipes. Operational factors include parameters related to network operation, such as water properties, pressure, or velocity. Finally, environmental factors are external conditions that affect network performance, such as climate, soil erosion, land use, etc.

The importance of physical factors in future fractures has been strongly demonstrated. Fares and Zayed [35], after consulting more than 20 experts, concluded that age had the greatest impact on failure risk, followed by material and failure rate. Meanwhile, Christodoulou et al., consider the age of the pipeline as an output variable called "lifetime". Materials are treated differently [29], with some authors only studying a certain material, while others consider all the different materials in the water network [36]. Several studies

have shown that pipes with smaller diameters tend to suffer more failures [37]. Regarding pipe length, longer lengths are assumed to be more exposed to failure risk [31]. Another way to extend the life of a pipe section might be to add some type of protection. In fact, protection-type methods have been shown to significantly extend the service life of pipelines, especially cathodic protection in iron pipes [38].

For the entire network, operational factors such as water pressure or flow rate are more difficult to obtain. Of all these factors, the number of previous failures (NOPF) is the most commonly used factor, and its influence on the occurrence of new pipeline failures has been widely documented. Pipes that are already broken are more prone to new breakage. Oliveira et al., used density-based spatial clustering analysis to conclude that poor fracture repair may generate new fractures similar to the previous one [39]. There are several points of view regarding water pressure. On the one hand, Shirzad et al., proposed that the performance of two prediction models, an artificial neural network and support vector regression, was improved when the hydraulic mean value was used as an input variable [40]. On the other hand, Jafar et al., defended the main effect of greater-than-average pressure fluctuations [41].

Data related to pipeline environments are uncommon, and these factors are sometimes estimated by the area under certain assumptions, for example, using pipeline clusters by location and historical fractures. Some scholars included traffic variables in their research and concluded that pipes under roads with heavy traffic are more likely to rupture than pipes under sidewalks with light traffic. Others have also studied the relationship between pipeline failure and soil liquefaction because fracture behavior changes if an earthquake occurs in the region. Fares and Zayed argued that the best parameter for proper soil classification is corrosivity [35]. Corrosion is an electrochemical phenomenon between two materials in contact with each other, resulting in the deterioration of the parent material. Since it is difficult to obtain this parameter directly, the soil type is used to express the approximate value of its corrosivity. There are many factors that affect corrosivity, and soil pH is considered a good indicator because corrosion occurs within a certain pH range.

When the data recorded are annual data, it is not possible to study the changes experienced by certain parameters over the course of a year. The authors have used the shorter time period to process the data to justify the importance of seasonal variation in pipeline ruptures. An important cause of failure is the water renewal time in the pipe, usually extended during the dry period. The longer the update time, the more glitches there will be. Also, in winter, when there is often heavy snowfall, pipes tend to suffer more failures due to the extra load the pipes have to withstand. Table 1 summarizes the main factors used in several studies over the past decade, including physical, operational, and environmental characteristics of three types.

**Table 1.** Statistical arrangement of risk factors and types.

| Type | Factor | English Description |
|---|---|---|
| | Pipe age | - |
| | Material | Pipe material |
| | Pipe diameter | - |
| Physical factors | Pipe length | - |
| | Pipe pressure | - |
| | Flow | Measuring point flow |
| | NOPF | Number of previous failures |
| | Soil type | Soil type |
| Environmental factors | Regional Environment | Area type |
| | Temperature | Temperature |
| | Type of damage | Failure type |
| Operational factors | Transportation | Traffic |
| | Other | Others |

### 3. Principle of the Random Forest Method

Ensemble learning is currently a widely utilized machine learning approach. Instead of being a singular machine learning algorithm, it is achieved by developing multiple models based on a dataset and integrating their respective results. Ensemble algorithms can be categorized into three types: bagging, boosting, and stacking. The fundamental concept of bagging involves constructing multiple independent estimators and determining the ensemble estimator's outcomes by employing the average or majority voting principle for their predictions. Random forests serve as a prominent example of the bagging approach.

Suppose there are $N$ samples and $M$ features. It has two meanings for "random": one is to randomly select samples and the other is to randomly select features. For each tree, training samples are randomly selected with the replacement method (such as $2N/3$) as the training set, and then $m$ features are randomly selected with the replacement method as the basis for the branching of the tree ($m << M$).

For each tree, there are $m$ features. To know whether a feature plays a role in this tree, the value of this feature can be changed randomly so that the contribution rate of this feature in this tree is zero. And then, the error rate of the test set before and after the change is compared, and the difference in the error rate is regarded as the importance of the feature in the tree. The test set refers to the remaining samples (out-of-bag samples) after $2N/3$ samples are extracted from the tree. The error caused by the out-of-bag samples as the test set is called out-of-bag error.

If $m$ features are calculated once in a tree, the importance of $m$ features in the tree can be calculated, but this can only represent the importance of these features in the tree, not the importance of features in the entire forest. Each feature appears in multiple trees, and the average value of the importance of this feature in multiple trees is the importance of the feature in the forest. As shown in Equation (1):

$$MDA(A_i) = \frac{1}{ntree} \sum_{t=1}^{ntree} (errOOB_{t1} - errOOB_{t2}) \tag{1}$$

where *ntree* represents the number of times the feature appears $A_i$ in the forest. $errOOB_{t1}$ represents the out-of-bag error after the attribute value $A_i$ in the $t$-th tree is changed, and $errOOB_{t2}$ represents the out-of-bag error of the normal value $A_i$ in the $t$-th tree, as shown in Figure 1.

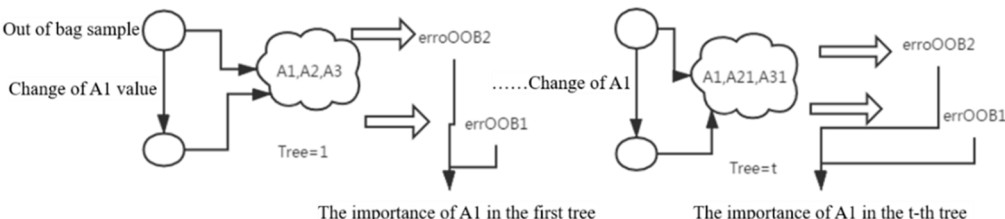

**Figure 1.** Iterative diagram of random forest.

In this way, the importance of all features in the forest is obtained. All features are sorted according to their importance, and some features with low importance in the forest are removed to obtain a new feature set.

Multiple iterations were carried out according to the above steps, gradually the relatively poor features were removed, and a new forest was generated each time. Stop the iteration until the number of remaining features is $m$. The $OOB$ index is used to evaluate the error rate of the out-of-bag samples in the forest. These samples are compared with the real value after the results of all samples are predicted, and the out-of-bag error rate of the forest can be obtained. The forest with the smallest out-of-bag error rate is selected as the final random forest model.

### 4. Application of Random Forest Method in Municipal Pipeline Network: Pipeline Network in Suzhou Industrial Park

*4.1. Data Preprocessing*

Urban utility network systems encompass a multitude of intricate data types, including spatial data, real-time operational data, maintenance and GPS dynamic data, equipment details, user information, and statistical data. Data preprocessing holds significant importance in predictive algorithms, primarily entailing the transformation of categorical data into numerical data, the analysis of abnormal situations, the management of missing and outlier values, and the standardization of diverse factors.

Since categorical data such as pipe material, pipe length, pipe pressure or flow needs to be converted to numbers, all factors can be analyzed graphically and descriptively by statistical measures to detect anomalies such as missing or abnormal values. Samples showing anomalies can be removed, but relevant information about the remaining variables can be lost. Therefore, both missing and abnormal values are filled with the median value of the factor. Finally, all factors are normalized to unify their format and scale. The main steps of data preprocessing: data cleaning, data integration, data transformation, and data reduction.

The records of damage characteristics are mostly textual descriptions, such as "Water Seepage at the West Side of Building 1, Z Company, X Street, Y Road". In order to accurately locate the damaged pipe section, word segmentation operation is considered for the fields in the destruction feature: "Water Seepage/West Side/Building/1/Z/Company/X/Street/Y/Road". The "jieba" word segmentation is the most popular word segmentation tool. The "jieba" word segmentation process mainly involves the following algorithms [42]:

(1) Implement efficient word graph scanning based on the prefix dictionary, and a directed acyclic graph (DAG) composed of all possible word formations characters is generated in one sentence;
(2) Using dynamic programming to find the maximum probability path, and find the maximum segmentation combination based on word frequency;
(3) For words not previously entered, HMM model based on word forming ability is adopted, and the Viterbi algorithm is used for calculation;
(4) Part-of-speech tagging based on the Viterbi algorithm;
(5) Extract keywords based on tf-idf and text rank models.

*4.2. Balancing Data and Exploratory Analysis*

This article mainly starts from the data point of view and uses the corresponding Python library (imblearn) to deal with unbalanced data sets. The SMOTE algorithm [43] is an improved method based on the random oversampling algorithm. Its basic idea is to analyze the minority class samples and artificially synthesize new samples based on the minority class samples and add them to the data set. As shown in the figure below, the algorithm flow is as follows:

1. For each sample $x$ in the minority class, calculate the distance between the point and other sample points in the minority class, and get the nearest $k$ neighbors (that is, perform the KNN algorithm on the minority class points);
2. Set a sampling ratio according to the sample imbalance ratio to determine the sampling magnification. For each minority class sample $x$, randomly select several samples from its $k$ nearest neighbors, assuming that the selected neighbor is $x'$;
3. For each randomly selected neighbor $x'$, construct a new sample with the original sample according to the following formula:

$$x_{new} = x + rand(0,1) \times (x' - x) \tag{2}$$

As can be seen from Figure 2, the proportion of majority class samples is much larger than that of minority class samples in the original data set. When synthesizing a new dataset through the SMOTE algorithm, it acts on the minority class samples, so that there is

a serious overlap between the original corrupted data and the synthetic data. The original data set contains 5376 samples, of which the sample size of the minority class is 55, the sample size of the majority class is 5321, and the ratio of positive and negative classes is about 1:100; the final synthetic dataset contains 6687 samples, of which the minority class sample capacity is 1645, the majority class sample capacity is 5042, and the ratio of positive and negative classes is about 1:3, which effectively improves the balance of unbalanced datasets. The risk assessment analysis is performed based on the improved dataset.

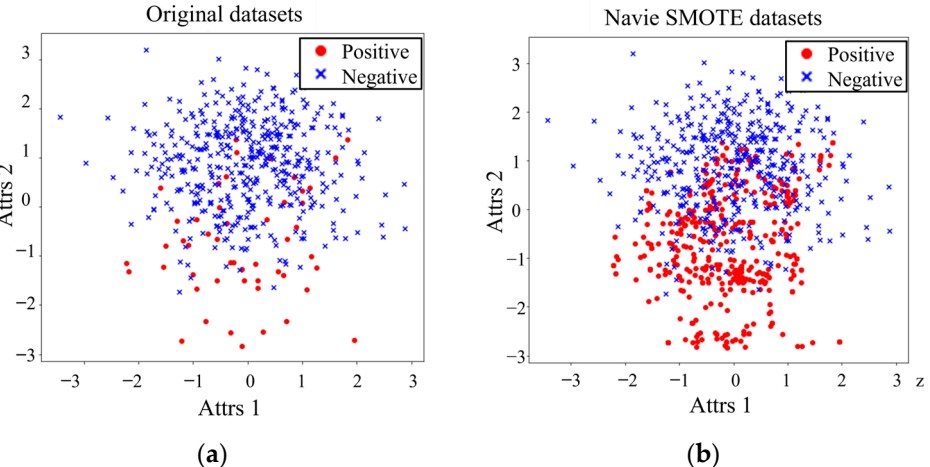

**Figure 2.** Comparison of SMOTE algorithm datasets. (**a**) Original datasets; (**b**) SMOTE datasets.

Exploratory data analysis (EDA) [44] refers to a data analysis method that explores the structure and regularity of the data by means of graphing, tabulating, calculating features, etc. based on the existing data. EDA focuses on the general description of data, which is not limited by models and scientific research assumptions.

As shown in Figure 3a, the distribution of pipes with Dia below 300 mm is relatively concentrated, accounting for 63.6% of the total. Figure 3b shows that the service life of most pipes is less than 25 Age, and the pipe age of 10~16 Age accounts for a large proportion, about 65.4% of the total. The total number of pipes is 75,615, of which ductile iron pipes account for 82.1%, steel pipe pipes account for 10.5%, PE pipes account for 5.7%, and PCCP pipes account for 1.1%. The longest pipeline length is 1142 m, and the minimum length is 1m; the minimum pipeline diameter is 25 mm, and the maximum diameter is 2200 mm; the maximum pipeline age is 26, and the minimum pipeline age is 1. The monitoring period of the pipeline is 7 months. Other data can be viewed in the attached table (Tables S1–S3).

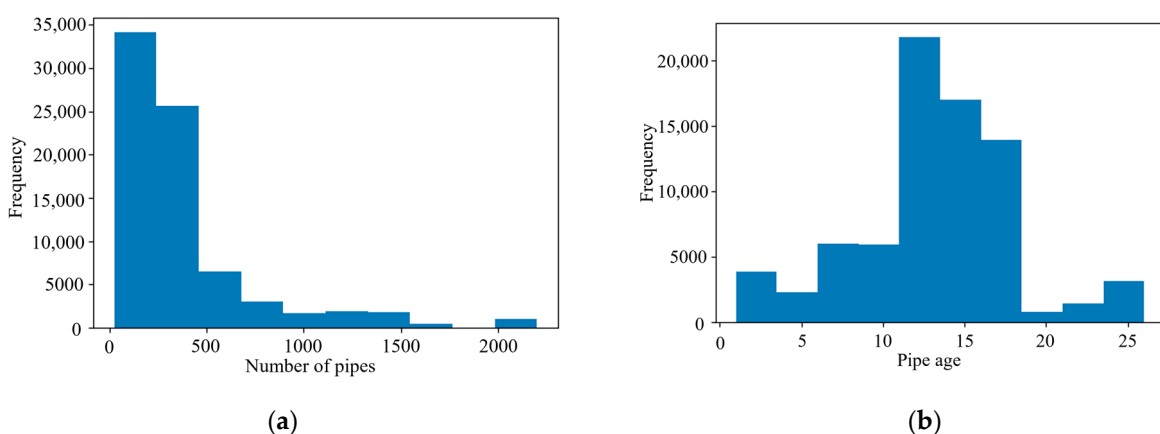

**Figure 3.** Frequency diagram of pipe diameter and pipe age distribution. (**a**) Pipe diameter; and (**b**) pipe age.

### 4.3. Optimization of the Random Forest Model

Hyperparameters are parameters whose values need to be preset before learning, and algorithms are rarely hyperparameter-free. For machine learning algorithms such as random forests, there are several hyperparameters that have a significant impact on the predictive accuracy of the model. Therefore, reasonable tuning of these hyperparameters, that is, hyperparameter optimization, is very important.

Ideally, and given enough data, it is best to randomly split the dataset into three parts: training, validation, and testing. The training set is used to fit the model; the validation set is used to estimate the prediction error for model selection; and the test set is used to evaluate the generalization error of the final mode. Since the data is usually small (here only 2400 data points), it is common that the generalization performance of the model cannot be truly reflected. In order to avoid biases in data selection, this study employs one of the most widely used validation methods for hyperparameter tuning and model evaluation processes, such as k-fold cross-validation (*k*-fold CV) [45,46]. In *k*-fold CV, the original sample *D* is randomly partitioned into *k* similarly-sized, mutually exclusive subsets, i.e., $D = D_1 \cup D_2 \cup \cdots \cup D_k$, and $D_i \cap D_j = \Phi (i \neq j)$. Each subset $D_i$ aims to maintain the consistency of data distribution to the greatest extent possible, beginning with stratified sampling of *D*, and subsequently using the union of *k*-1 subsets as the training set, while the remaining subset serves as the test set. This approach results in *k* sets of training/testing sets, facilitating *k* training and testing iterations. Although no strict rules exist for determining the value of *k*, *k* = 10 (or 5) is commonly utilized in the machine learning field. In this context, following the recommendations of Kohavi (1995) [8] and Wong (2015) [10], the number of folds *k* is set to 10, which is associated with the trade-off between computation time and bias. As a result, this study employs a 10-fold CV method for model validation (an example of which is depicted in Figure 4).

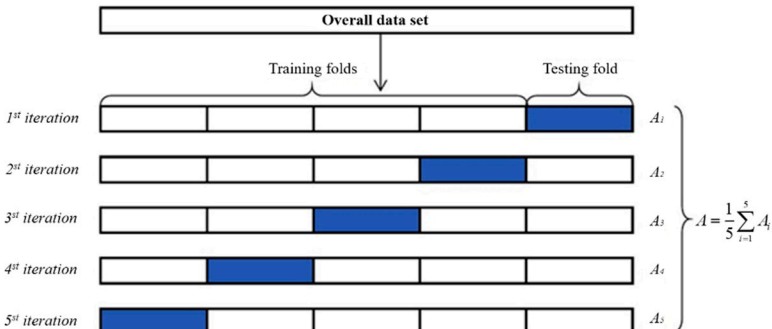

**Figure 4.** Cross-validation.

### 4.4. Comparison of Optimized Random Forest Model with Other Models

In this section, we provide a comprehensive comparison of the prediction accuracy and robustness of the hyperparameter-optimized random forest algorithm against three alternative machine learning methods, all under the framework of 10-fold cross-validation. As part of the study, we employ decision trees, logistic regression, and support vector machine algorithms, which are implemented utilizing the well-established Scikit-learn library. To ensure a fair comparison and control for extraneous variables, all three models being compared are subjected to hyperparameter tuning using the Grid Search method. This rigorous approach allows for a more accurate assessment of the performance of each model. The outcomes of the models, including relevant performance metrics, are presented in the table below, offering insights into the relative strengths and weaknesses of the various algorithms under consideration.

Upon examining the results in Table 2, it becomes evident that the accuracy of logistic regression and random forest models (0.77) is considerably greater than that of support vector machines and decision trees (0.72, 0.59). In addition, the recall rates for minority class samples in logistic regression (0.53) and random forest (0.59) models exhibit a sub-

stantial enhancement when contrasted with those of decision trees (0.49), support vector machines (0.46), and logistic regression (0.53). The ROC curves of the four models are drawn in Figure 5.

**Table 2.** Model evaluation results.

|  |  | Precision | Recall | Fl-Score | Support |
|---|---|---|---|---|---|
| Decision trees | 0 | 0.72 | 0.67 | 0.7 | 43 |
|  | 1 | 0.33 | 0.49 | 0.36 | 18 |
|  | Accuracy |  |  | 0.59 | 61 |
|  | Macro avg | 0.53 | 0.53 | 0.53 | 61 |
|  | Weighted avg | 0.61 | 0.59 | 0.6 | 61 |
| Logistic regression | 0 | 0.77 | 0.95 | 0.85 | 43 |
|  | 1 | 0.75 | 0.53 | 0.46 | 18 |
|  | Accuracy |  |  | 0.77 | 61 |
|  | Macro avg | 0.76 | 0.64 | 0.66 | 61 |
|  | Weighted avg | 0.77 | 0.77 | 0.74 | 61 |
| Support vector machines | 0 | 0.72 | 1.00 | 0.83 | 43 |
|  | 1 | 1.00 | 0.46 | 0.11 | 18 |
|  | Accuracy |  |  | 0.72 | 61 |
|  | Macro avg | 0.86 | 0.53 | 0.47 | 61 |
|  | Weighted avg | 0.80 | 0.72 | 0.62 | 61 |
| Random forest | 0 | 0.78 | 0.93 | 0.85 | 43 |
|  | 1 | 0.70 | 0.59 | 0.50 | 18 |
|  | Accuracy |  |  | 0.77 | 61 |
|  | Macro avg | 0.74 | 0.66 | 0.68 | 61 |
|  | Weighted avg | 0.76 | 0.77 | 0.75 | 61 |

Notes: 0 means majority class samples, 1 means minority class samples.

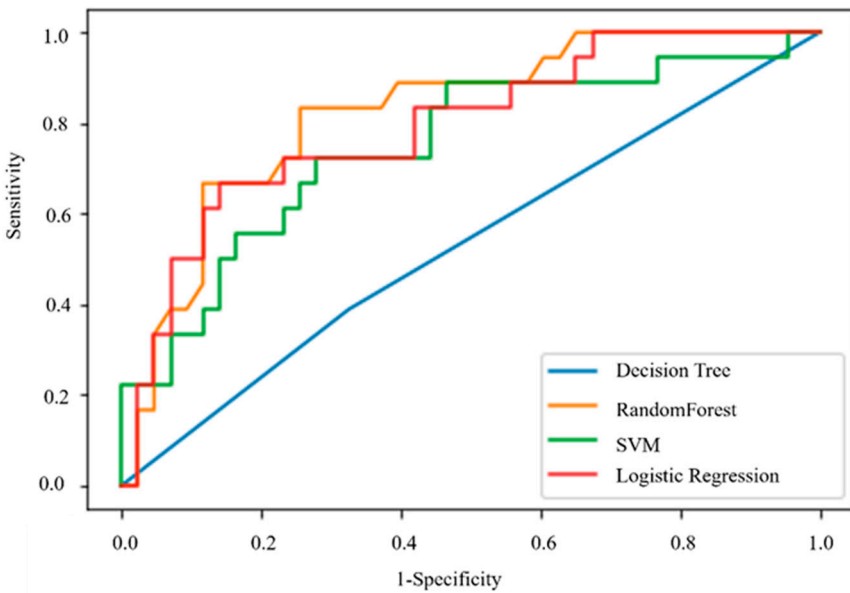

**Figure 5.** Comparison of ROC results of four models.

Upon comparison of the areas under the ROC curve, it becomes evident that the random forest attains the maximum AUC value (0.82), subsequently followed by logistic regression (0.79), support vector machine (0.74), and decision tree (0.53). These findings demonstrate that, within this particular dataset, the ensemble learning algorithm random forest delivers superior performance.

*4.5. Calculation Results of Random Forest Risk Assessment*

4.5.1. Univariate Analysis

Firstly, the age and damage of the pipe section are analyzed. The pipe section with a pipe age of more than 15a has a large proportion of damage, and the failure probability of the pipe section also shows an upward trend with the increase of the age of the pipe section, as shown in Figure 6.

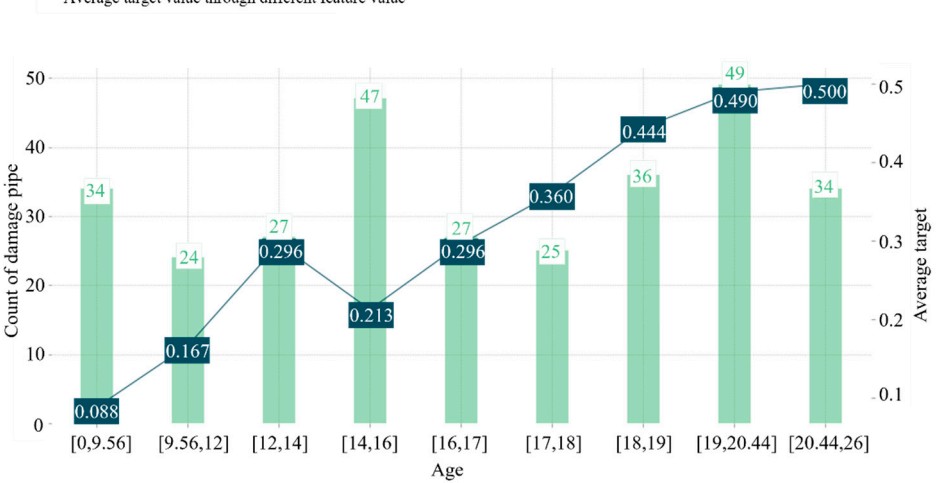

**Figure 6.** Pipe age and damage.

Then the violin diagrams of the pipe diameter, soil pH, and failure conditions are presented, respectively, as shown in Figure 7. The diameter distribution of vulnerable pipe sections is relatively scattered, and the distribution of undamaged pipe sections is relatively concentrated, both of which have relatively obvious discrete values. And the damage density of smaller pipe diameters is greater. The distribution of pipe section failure in soil pH is also concentrated in the middle and lower parts of the graph, indicating that the failure mostly occurs in the range of low soil pH.

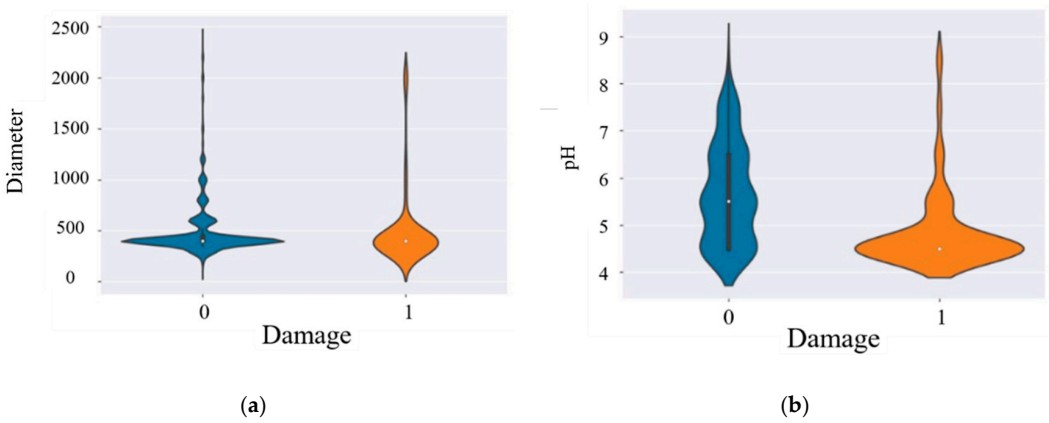

(**a**)                                    (**b**)

**Figure 7.** Violin diagram of the failure condition. (**a**) pipe diameter; (**b**) soil pH.

The soil pH range is further analyzed. It shows that the failure probability of the pipe section is 0.46 when the pH is in the range of 4.5 to 5.5, as shown in Figure 8.

4.5.2. Feature Interaction Analysis

The darker the color in the interaction analysis diagram, the greater the probability of failure of the pipeline, and the size of the circle indicates the number of pipelines here.

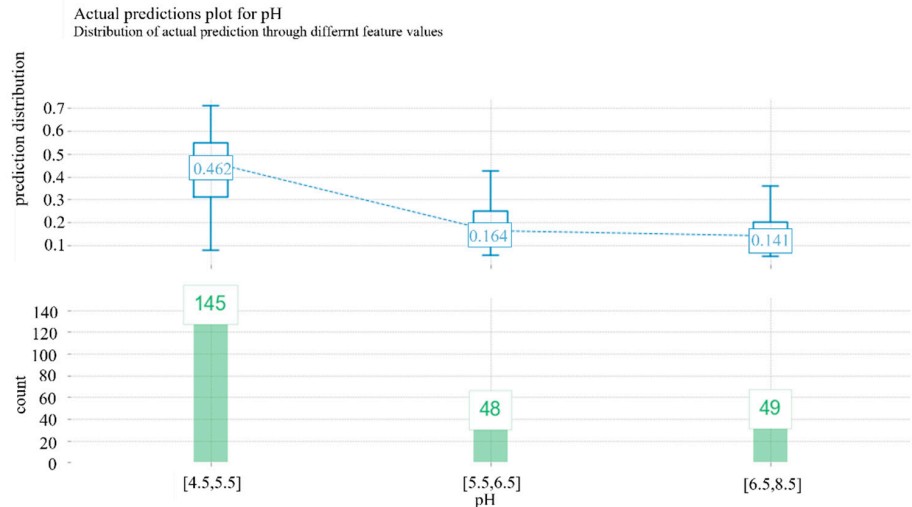

**Figure 8.** The proportion of soil pH distribution and failure.

Figure 9 shows the interaction analysis of soil pH and pipe age. It can be seen from the figure that the lower the soil pH, the greater the pipe age and the higher the damage risk. The risk of pipeline failure increases with lower pH and increases with age, so monitoring of pipelines in this area should be increased.

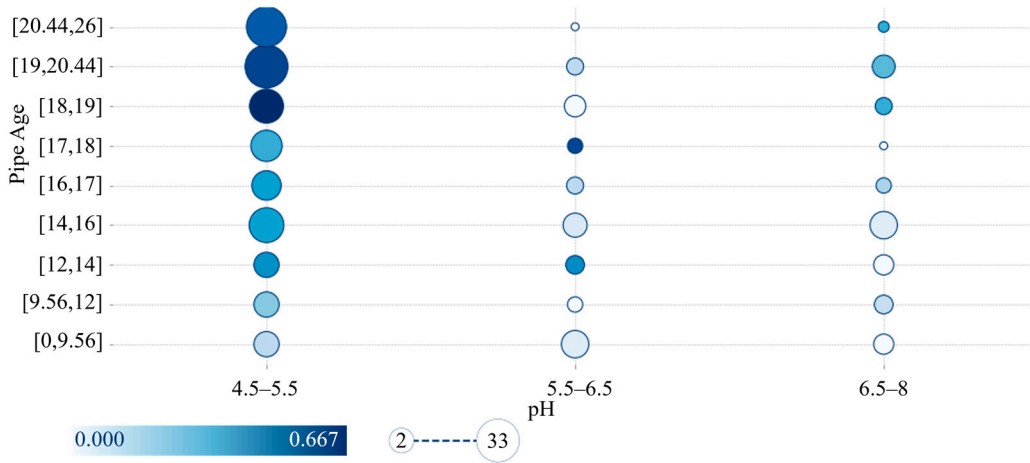

**Figure 9.** Interaction analysis of soil pH and pipe age.

Figure 10 (0 means flow, 1 means pressure drop) shows the interactive analysis results of pressure drop and flow. It can be seen from the figure that in the pipe section without pressure drop, the greater the flow, the easier it is to cause pipe section damage. In the section where pressure drop occurs, the effect of flow on damage is negligible, because the distribution of flow is more dispersed. Since the color depth on the left side of the figure is stronger, it can be inferred that the impact of pressure drop is greater than that of flow.

### 4.5.3. Feature Importance and Decision Impact

Table 3 shows the feature importance and ranking of pipeline risk index factors. It can be seen from the table that the previous failure (NOPF) has the greatest impact on the damage of the pipe section, which can be understood as the secondary damage probability of the pipe section being very large. It may be that the damaged and repaired pipeline can easily cause stress concentration, and it is easily damaged again under ununiform pressure drops. Among the pipes, steel pipes have the greatest influence. Compared with ductile iron and PE pipes, steel pipes have lower corrosion resistance, and are suitable for a

small amount of application in non-key parts. Ductile iron pipes account for the largest proportion in Suzhou Industrial Park, about 80%, steel pipes account for about 10%, and PE pipes account for about 5%. In addition, soil pH, pipe age, and pressure drop are also important influencing factors.

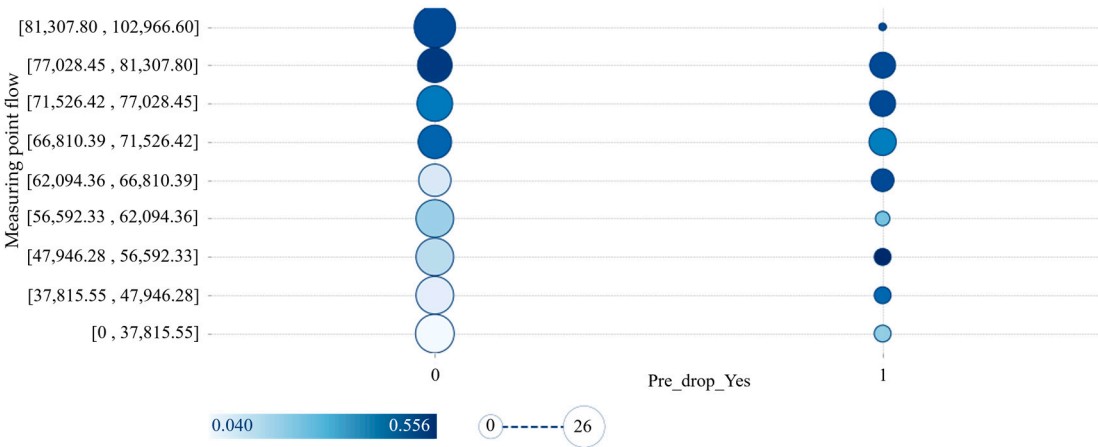

**Figure 10.** Interactive analysis of pressure drop and flow.

**Table 3.** Importance ranking of features.

| Weights | Feature | Weights | Feature |
|---|---|---|---|
| 0.3753 | NOPF_More | 0.0022 | Pre_drop_No |
| 0.1581 | Mat_Steel Pipe | 0.0009 | Mat_Ductile Iron |
| 0.1543 | pH | 0.0001 | Pipe pressure |
| 0.1166 | Pipe age | 0.0001 | Flow |
| 0.0750 | Pre_drop_Yes | 0.0001 | NOPF_No |
| 0.0425 | Pipe diameter | 0.0001 | Mat_PE |
| 0.0319 | Pipe length | | |
| 0.0282 | NOPF_1 | | |
| 0.0150 | Mat_Other | | |

A partial Dependency Picture (PDP) shows the dependencies between a target response and a set of "target" features and marginalizes the values of all other features ("complementary" features). Intuitively, partial dependencies can be interpreted as the expected target response as a function of the "target" feature. The sub-features are the partial dependencies of each feature value and the partial dependencies of interactive features based on test results displayed through the PDP. It is important to note that PDP assumes that target features are independent of complement features, and this assumption is often rebutted in practice. Therefore, the PDP here is only used as a trend analysis rather than a quantitative analysis.

For the characteristics of the pipe material, it can be seen from Figure 11 that when the pipe is replaced by another pipe from ductile iron, the target response value increases, which proves that the risk of damage to the pipe section increases. It shows that the pipes using ductile iron pipes are more durable than pipes using other pipes, which will cause this phenomenon. Therefore, ductile iron pipes should be used in places that are prone to damage to water pipes to reduce damage to the pipeline network.

For the NOPF, as shown in Figure 12, the pipe segment is transformed from the previous failure state (NOPF_No_0) to the non-previous failure state (NOPF_No_1), and the predicted probability will decrease, indicating that the risk of the pipe segment is reduced. For pipelines with previous failures and pipelines without previous failures, the corresponding value of the PDP of pipelines with previous failures increases, indicating

that the previous failures are easy to cause damage to the pipeline, and the monitoring of this pipeline should be strengthened.

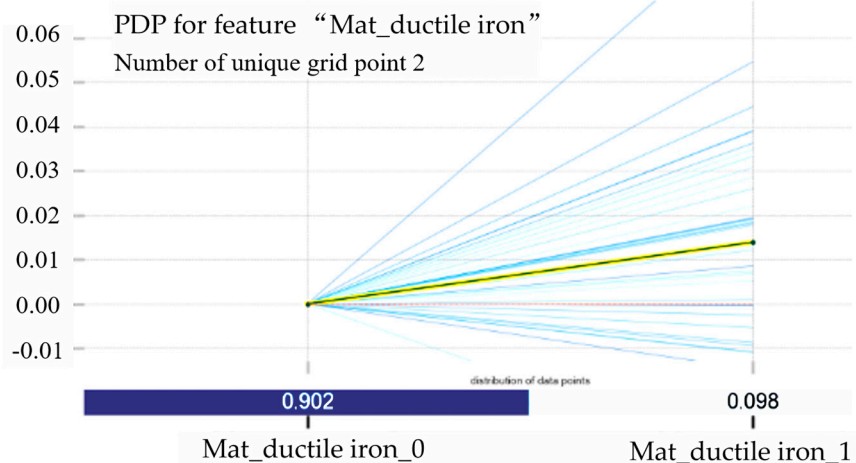

**Figure 11.** The PDP of ductile iron pipe.

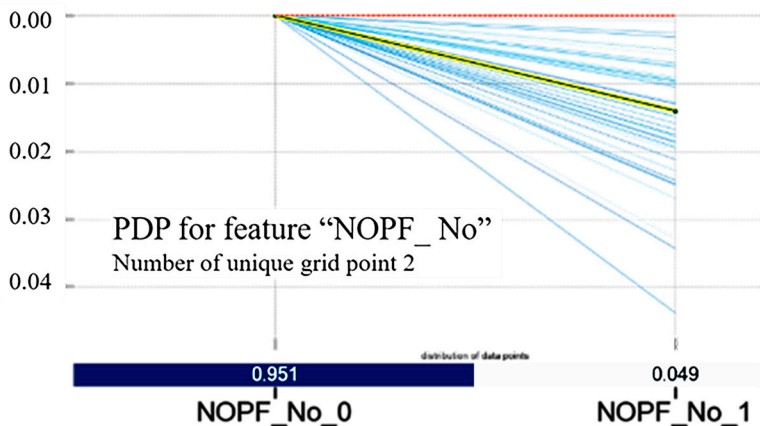

**Figure 12.** The PDP of the pipe segment NOPF.

The risk probability will increase when the pipe material is changed from non-steel pipe to steel pipe under the same pipe age. As the pipe age increases, the risk probability also increases under the same pipe material conditions. The interactive effect results of the PDP are shown in Figure 13, which shows the relationship between the risk probability and the joint value of steel pipe or not and pipe age, and the interaction between these two features can be clearly seen from the figure. For the case of pipe age less than 10 years, the risk probability increases slowly with the change of pipe materials, while for the case of pipe age more than 10 years, there is a strong increase.

4.5.4. Feature Contribution and Decision Paths

SHAP is based on Shapley values. Shapley values are a concept in game theory. What Shapley does is quantify the contribution of each feature to the prediction made by the model, that is, explain this prediction, and show how each feature contributes to the prediction. SHAP values assign the attributed value for each feature to the expected change in the model's prediction when that feature is adjusted.

A pipe segment is arbitrarily selected (idx = 120), and a SHAP (Shapley value) plot is generated to illustrate the prediction results for this specific segment, as depicted in Figure 14. In this visualization, red bars represent features that positively contribute to the prediction, while blue bars indicate features that negatively contribute. The length

of the red bars signifies the magnitude of the positive contribution of the corresponding feature to the risk prediction probability for the pipe segment, given the data at hand. Furthermore, the plot provides a quantitative representation of the contribution values for each feature. For the three parameters with previous pressure, pipe length, and more number of previous failures, the random forest model is more sensitive to these three parameters than without the number of previous failures and pipe diameter. It shows that having previous pressure, pipe length, and more number of previous failures have a positive contribution to the failure of the pipe, which may be because the pipe with previous pressure will accelerate the fatigue failure of the pipe; the pipe with long pipe length has a larger surface area, so there may be more localized corrosion, scaling or other defects; secondly, the pipeline that has failed previously may already have some degree of damage, corrosion or material deterioration.

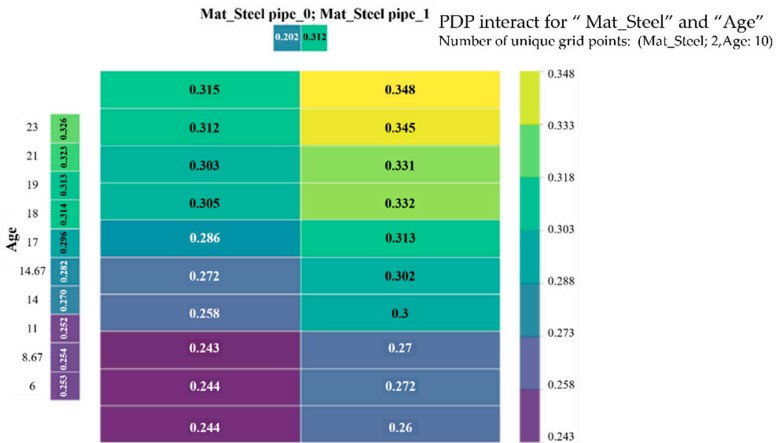

**Figure 13.** Interaction analysis diagram of pipe material and pipe age PDP.

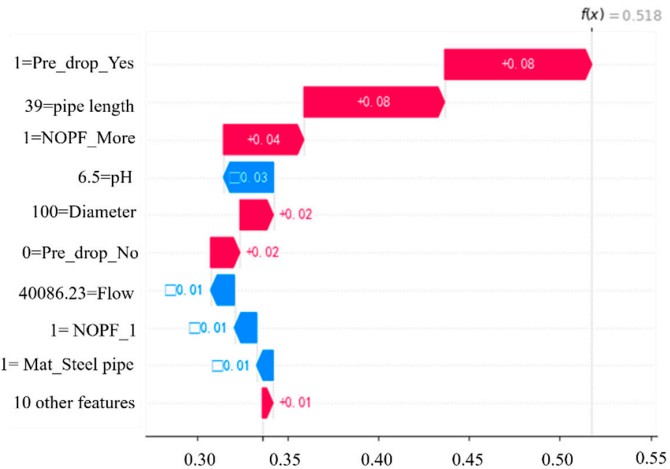

**Figure 14.** idx = 120 pipe segment SHAP.

## 5. Conclusions

Based on the monitoring data of the municipal water supply network in Suzhou Industrial Park, a random forest algorithm model for risk prediction of the municipal water supply network based on machine learning was established, and the single variable and variable were analyzed to obtain the feature importance and contribution rate. Improved model interpretability. The main conclusions are as follows:

(1)  Among several machine learning algorithms, SVM, Naive Bayes, and Decision Trees do not fit the training set as well as Logistic Regression and Random Forest. The accuracy rate of logistic regression and the random forest is significantly higher than

that of the support vector machine and decision tree, and for minority sample logistic regression, the recall rate of random forest is also significantly higher than decision trees and support vector machines;

(2) The comparison of the area under the ROC curve shows that the AUC value of the random forest is the largest (0.82), followed by logistic regression (0.79), support vector machine (0.74), and decision tree (0.53). It can be seen that the random forest of the integrated learning algorithm performs best on this data set;

(3) Among the pipes, steel pipes have the greatest influence. Compared with ductile iron and PE pipes, steel pipes have lower corrosion resistance, so they are suitable for a small amount of application in non-critical parts. The previous failure (NOPF) has the greatest impact on the damage to the pipe section. It can be understood that the secondary damage probability of the pipe section is very high, and the damaged and repaired pipe is likely to cause stress concentration and damage again under uneven pressure drops;

(4) Through further analysis of the municipal pipe network risk assessment model under the established random forest algorithm, the influence of every single variable and pairwise interactive variables on the damage of pipe sections is given; through Feature Importance, Permutation Importance) to measure the importance of each feature in the data set, and the quantitative relationship between the feature and the final prediction result is given by the feature contribution (SHAP). Taking a single sample as an example, the decision-making process of the data is shown.

**Supplementary Materials:** The following supporting information can be downloaded at: https://www.mdpi.com/article/10.3390/w15101964/s1, Table S1: total; Table S2: chlorine; Table S3: turbidity.

**Author Contributions:** H.C.: conceptualization, methodology, software, writing—original draft. D.H.: methodology, project administration, formal analysis. Q.L.: visualization, data curation. Z.Z.: supervision, investigation, funding acquisition. A.T.: review and editing. All authors have read and agreed to the published version of the manuscript.

**Funding:** This research was supported by the National Natural Science Foundation of China (No. 41672287, 51778197), the Hainan Province Key R&D Program (Social Development) Project of China (No. ZDYF2022SHFZ089). and the Jiangsu Province Key R&D Program (Social Development) Project of China (No. BE2021681). The supports are gratefully acknowledged.

**Data Availability Statement:** All data, models, and code generated or used during the study appear in the published article.

**Conflicts of Interest:** The authors declare that they have no known competing financial interest or personal relationship that could have appeared to influence the work reported in this paper.

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
