# Peer review of "Application Research on Risk Assessment of Municipal Pipeline Network Based on Random Forest Machine Learning Algorithm"

_water, doi:10.3390/w15101964_

Round 1

Reviewer 1 Report

Pipe failures in urban water and sewer systems are a global concern for water utilities. Pipe failure prediction models provide important support for understanding pipe failure behavior and managing pipe failure risk. The subject of this manuscript is of interest. Nevertheless, the quality of the manuscript needs to be significantly improved for publication. The main comments are as follows.

(1) The novelty of this work needs to be stressed. Random forest method, as well as some other machine learning methods, is a commonly used method in this area. What are the advantages of this approach over other algorithms?

(2) The properties and operation environment are significantly different between urban water supply system and urban drainage system, resulting in different pipe failure mechanisms between them. Why did this work put them together for model development?

(3) The data used in this work were not sufficiently described.

(4) Too many pages and figures in this manuscript. The manuscript could have been more concise.

(5) Too much editing typos and grammatical issues in the manuscript. In the abstract, for example, the subject is missing from line 24 to line 34.

(6) The references sequential number of some references are incorrect, e.g. Ref. [39] in Line 88, Ref. [17] in Line 122. Refs [12,13] are not cited in the text.

(7) There are too many acronyms in the figures, which were not defined, e.g. the acronyms in Figure 16.

(8) pH should not be written as PH.

The quality of English needs to be significantly improved. 

Author Response

Dear Reviewers:

Thank you for your letter and for the reviewer's comments concerning our manuscript entitled “Application research on risk assessment of municipal pipeline network based on random forest machine learning algorithm” (ID: water-2378527). Those constructive comments are all valuable and very helpful for revising and improving our papers well as the important guiding significance to our research. We have studied the comments carefully and have made corrections which we hope meet with approval. We have added line numbers to the revised manuscript to facilitate the search for specific revised positions. And “Track Changes” versions are also submitted for comparison of changes before and after.

Reviewer #1: Pipe failures in urban water and sewer systems are a global concern for water utilities. Pipe failure prediction models provide important support for understanding pipe failure behavior and managing pipe failure risk. The subject of this manuscript is of interest. Nevertheless, the quality of the manuscript needs to be significantly improved for publication. The main comments are as follows:

  1. The novelty of this work needs to be stressed. The random forest method, as well as some other machine learning methods, is a commonly used method in this area. What are the advantages of this approach over other algorithms?

Response: Thanks to the reviewers for their careful review and suggestions, and agreed to emphasize the innovation of the work and explain the advantages of the random forest method. Added instructions for this section on lines 104 to 115 and highlighted the text in yellow.

  1. The properties and operation environment are significantly different between urban water supply systems and urban drainage system, resulting in different pipe failure mechanisms between them. Why did this work put them together for model development?

Response: Thanks to the reviewers for their careful review and feedback, pointing out the differences in the nature and operating environments of urban water supply systems and urban drainage systems. Due to the my negligence, drainage is included, and the actual research is only for water supply pipes. Water supply pipelines protect people's livelihood in municipal pipelines and are the most important part of lifeline projects.  We have dealt with the part about drainage content in the article. We made amendments at line 15 of the abstract, at line 35 of section one, and at lines 484 and 486 of section five. Finally, thanks to the reviewers for their pointers.

  1. The data used in this work were not sufficiently described.

Response: Thanks a lot for reminding us of this important point. This kind of problem occurred because the author did not fully and carefully describe some data in the figure and text. Therefore, the following modifications were made to the paper: in line 472 to 480.

  1. Too many pages and figures in this manuscript. The manuscript could have been more concise.

Response: Thanks a lot for reminding us of this important point. We quite agree with your comment. Your comments are very important to the improvement of our paper. As you pointed out, we deleted Figure 16 in Section 4 of the paper, thinking that the information shown in this figure cannot have a certain effect on the conclusion of the paper. Finally, thank you again for your careful review.

  1. Too much editing typos and grammatical issues in the manuscript. In the abstract, for example, the subject is missing from line 24 to line 34.

Response: Thanks a lot for reminding us of this important point. We quite agree with your comment. I have changed the content of the original lines 22 to 24 to the following:

Finally, the risk probability associated with each pipe segment sample is obtained, the risk factors affecting the pipe segment's failure are determined, and their relevance and importance ranking is established.

  1. The references sequential number of some references are incorrect, e.g. Ref. [39] in Line 88, Ref. [17] in Line 122. Refs [12,13] are not cited in the text.

Response: Thanks a lot for reminding us of this important point. We quite agree with your comment. Because my negligence caused this kind of problem, some reference ordering problems in the paper have been revised. All the problems of ordering references and format of references in this paper have been solved. Finally, thank you again for your careful review.

  1. There are too many acronyms in the figures, which were not defined, e.g. the acronyms in Figure 16.

Response: Thanks a lot for reminding us of this important point. Your suggestions for modification are crucial to the improvement of this article. Due to negligence in writing and self-review, We explain the meaning of the abbreviations in some pictures. We re-examined and revised the following places:

  • Table1: Dia (pipe diameter)
  • Table1: Age (pipe age)
  • Table1: Material (Pipe material)
  • Table1: Len (Pipe length)
  • Table1: Pre (Pipe pressure)
  • Table1: Flow (Measuring point flow)
  • Table1: NOPF (Number of previous failures)
  • Line 390: (0 means flow, 1 means pressure drop)
  • Line 424: the previous failure state (NOPF_No_0)
  • Line 425: the non-previous failure state (NOPF_No_1)
  1. pH should not be written as PH.

Response: Thanks a lot for reminding us of this important point. Your suggestions for modification are crucial to the improvement of this article. All references to PH herein have been changed to pH.

Reviewer 2 Report

The paper has potential to meet publication standards but needs a lot of work yet. The English language usage is generally quite good but there are many areas where improvement in grammar is necessary and better explanation of the findings and the validity thereof

Many issues where the grammar being used leaves considerable uncertainty as to what is being discussed

Author Response

Dear Reviewers:

Thank you for your letter and for the reviewer's comments concerning our manuscript entitled “Application research on risk assessment of municipal pipeline network based on random forest machine learning algorithm” (ID: water-2378527). Those constructive comments are all valuable and very helpful for revising and improving our papers well as the important guiding significance to our research. We have studied the comments carefully and have made corrections which we hope meet with approval. We have added line numbers to the revised manuscript to facilitate the search for specific revised positions. And “Track Changes” versions are also submitted for comparison of changes before and after.

Reviewer #2: The paper has potential to meet public on standards but needs is a lot of work yet. The English language usage is generally Good, but there are many areas where improvement in grammar is needed

Major issues include:

1) There is confusing language used in description of pipeline' versus urban’ A pipeline in English refers to a major pipe which conveys liquid of some type, from location to another location whereas 'urban' water distribution pipes are very interrelated -e.g. perhaps 10,000pipes carrying water to different parts of a city, with many connections, and highly interrelated such that, if a pipe is damaged/broken, then the there are other means by which the water can be distributed to individual buildings (i.e. as a means of ensuring better connectivity of pipes to water users. There is continuing confusion in the existing form of the paper as to what is claimed, from the analyses undertaken. The utility of the method is greatly influenced by whether the analyses are truly pipelines or a network - at this point, it is very unclear.

Response: First of all, thank you very much for your comments. We quite agree with your comment. We thank the reviewers for their careful review and feedback, pointing out possible confusion in the article regarding the use of the concepts of pipeline and city. Acknowledging that the article may have some ambiguity in this aspect, I re-examined the article and clarified that this article only focuses on water supply pipelines, because water supply pipelines are the most important part of lifeline projects in the municipal pipe network to protect people's livelihood, to ensure that the use of concepts is more in line with the actual situation.

2) Drainage pipes in a network don't have a full flow, even to the point of one. Can measure pressure in pipes that don't flow fully. What does that do in the provided text.

Response: Thanks a lot for reminding us of this important point. We quite agree with your comment. Indeed, the pipes in a sewer network are not always full of flow. This is an important practical factor that needs to be considered in my research. For this proposal, We makes a detailed description in Line 69 to 71 and in Line 175 to 180 of the second section.

3) The abstract is written as a review of what is in the paper. Instead, an abstract should be indicating the primary results that will be discussed in the paper. Hence, the abstract needs to be totally re-written.

Response: Thanks a lot for reminding us of this important point. We quite agree with your comment. The new summary:

Urban municipal water supply and drainage pipes is an important part of underground pipelines, and their scale continues to expand. Due to the continuous improvement in the quality and quantity of data available for pipeline systems in recent years, traditional pipeline network risk assessment cannot cope with the improvement of various monitoring methods. Therefore, this paper proposes a machine learning-based risk assessment method for municipal pipe network operation and maintenance and builds a model example based on the data of a pipeline network base in a park in Suzhou. We optimize the random forest learning model, compare it with other centralized learning methods, and finally evaluate the model's learning effect. Finally, the risk probability associated with each pipe segment sample is obtained, the risk factors affecting the pipe segment's failure are determined, and their relevance and importance ranking is established. The results show that the most influential factors are pipe material, soil properties, service life, and the number of past failures. The random forest algorithm demonstrates better prediction accuracy and robustness on the dataset.

4) The nature of data being measured, and to be used in the AI work is vague. Some data refer to 'real time', and others are not e.g. 'soil' type.

Response: Thanks a lot for reminding us of this important point. We quite agree with your comment. We have submitted the relevant raw data Table S1, Table S2 and Table S3.

5)Reference is made to violin diagrams and SHAP but there is no differentiation between what one type of diagram versus another is being relied upon.

Response: Thank you for your careful review. We admit that the content of Figure 7 and Figure 15 is similar, so we will delete Figure 15.

6) Line 23 refers to established based on pipeline network data. To what is that referring?

Response: Thanks a lot for reminding us of this important point. We quite agree with your comment. We have revised the Introduction to outline the monitoring of the pipeline for the most part.

7)Line 42 refers to leak detection technology – needs clarification as to what that implies?

Response: Thanks a lot for reminding us of this important point. We quite agree with your comment. We have described and referenced the leak detection technique on lines 42 to 68 of the paper.

8) What types of sensors are being referred to in Line 51? – that influences the viability of what is being indicated as being solved. More specification on what is involved in multivariate monitoring is needed.

Response: Thanks to the reviewers for their careful review and feedback pointing out possible deficiencies in the description of sensor types and multivariate monitoring. Acknowledge that there may be deficiencies in the presentation in this area, and indicate that you will improve this in the revised manuscript. To solve this problem, We read the relevant literature and made amendments in lines 69 to 71.

9) Line 57 – statements are made about efficiency without any justification.

Response: Thanks a lot for reminding us of this important point. We quite agree with your comment. We are very sorry for the wrong citations of the figures. We are sorry that this part of the research has not been fully discussed, so some research on machine learning has been supplemented in lines 80 to 115 of Section 1. Finally, I sincerely thank you for pointing out the problems in this part.

10) Lines 67 – 69 the risk index system of pipeline network is still not perfect – This reviewer hash no idea what is being referred to.

Response: Thanks a lot for reminding us of this important point. We quite agree with your comment. We believe that this part of the content is not very accurate, so this part of the content has been deleted.

11) What is being referred to in line 106-107? An integrated database?

Response: Thanks a lot for reminding us of this important point. Yes, this is a database that integrates pipeline physical parameters (such as pipe length and age), related facilities (such as pumping stations and water plants), accident records (such as damage causes), monitoring technologies (such as fiber optic pressure gauges), etc.

12) Table 1 refers to many risk factors and types – some of those vary significantly from hour to hour, and some vary spatially. What is being assumed in the paper and the database that is available?

Response: Thanks a lot for reminding us of this important point. I am sorry that the reason for this part is not described in detail in the article. Therefore, a certain modification has been made to the section second, and the modified content is as lines 145 to 202.Finally, thank you again for your careful review.

13) What is the spatial frequency of real-time data at the various points the pipeline system?

Response: Thanks a lot for reminding us of this important point. For this question, we packaged the original data together with the revised manuscript and sent it to you.

14) Figure 2 needs better explanation and how the original corrupted data and the synthetic data needs to be explained.

Response: Thanks a lot for reminding us of this important point. We quite agree with your comment. We are very sorry for our negligence of this content. In addition, the original data set contains 5376 samples, of which 55 are minority samples and 5321 are majority samples, the ratio of the two is about 100:1. Modifications were also made in lines 305 to 308 in the manuscript. Thanks for your comment.

Minor issues include:

1) Grammar in line 81

Response: Thanks a lot for reminding us of this important point. We believe that the content of this part does not match the content described in this article, so this part of the content is deleted. Thanks to the reviewer for raising the grammatical issue of this sentence.

2) PH should be pH

Response: Thanks a lot for reminding us of this important point. Your suggestions for modification are crucial to the improvement of this article. All references to pH herein have been changed to pH.

3) Line 86 to 89 are unclear

Response: Thanks a lot for reminding us of this important point. We quite agree with your comment. Your comments are very important to the improvement of our paper. After careful reading, We found that the content of this part did not match the content of the full text, so this part of the content was also deleted and revised into lines 69 to 71 of the section first.

We also found a native English editor to modify the grammar and sentence structure in the article, which improved the language expression of the article. The attachment is an Editorial Certificate about this article.

Reviewer 3 Report

Although the aim of the work is evident, the Authors are not clear throughout their explanations: the manuscript, as it is, cannot be considered as a research paper, representing a mere application of standard machine learning methodologies. The Authors should give more emphasis on the objective and on the application procedure of the proposed methodology for real situations. 

To make the manuscript apt for publication, a thorough reorganization must be done. In particular, more references have to be included; the section in which material and methods are described has to be rearranged; original data must be reported (number of samples, characteristics, ...); the results obtained should be properly presented, together with more readable (and error-free) figures. The innovative character (if any) of the paper should also be properly described. In particular, in the first part of the paper the Authors prove the better performance of the random forest (and logistic regression) with respect to other methodologies. But, according to the results reported in Table 2, it appears that the predictive ability of all the methodologies is, in any case, not high enough, and could compromise real applications of pipe replacement planning and risk assessment. Such issues must be discussed in more detail by the Authors in their manuscript.

Other specific issues:

References should be integrated with more citations related to the subject of pipe replacement planning and pipe burst predictions. Moreover, bibliography is not up to date (actually, citations of any papers from years 2021, 2022 and 2023 are missing).

On page 2, line 88, ref. [39] has no meaning.

On page 6, Section 4.2, the functioning of the SMOTE algorithm is not clear: the Authors should give a detailed explanation or, at least, a useful reference for understanding it. Moreover, the Authors should also include the details of the original dataset: total number of samples and relative percentage of negative and positive ones.

Figure 4 on page 7 contains errors: 'interation' should be 'iteration'.

On page 11, the Authors have to describe the meaning of the interaction plots. Moreover they must explain the meaning of 'pressure drop': is it a pressure difference or a total head difference between pipe nodes? .

Figure 11 on page 12 contains errors in the x-axis. The Authors should better explain the meaning and the importance (if any) of the Partial Dependency Picture (PDP), illustrated in Figures 11 and 12.

Extensive English editing is required: many sentences miss the verb or are gramatically incorrect. Others are difficult to understand in their meaning.

Author Response

Dear Reviewers:

Thank you for your letter and for the reviewer's comments concerning our manuscript entitled “Application research on risk assessment of municipal pipeline network based on random forest machine learning algorithm” (ID: water-2378527). Those constructive comments are all valuable and very helpful for revising and improving our papers well as the important guiding significance to our research. We have studied the comments carefully and have made corrections which we hope meet with approval. We have added line numbers to the revised manuscript to facilitate the search for specific revised positions. And “Track Changes” versions are also submitted for comparison of changes before and after.

Reviewer #3:

Comments and Suggestions for Authors

Although the aim of the work is evident, the Authors are not clear throughout their explanations:

  1. The manuscript, as it is, cannot be considered as a research paper, representing a mere application of standard machine learning methodologies. The Authors should give more emphasis on the objective and on the application procedure of the proposed methodology for real situations.

Response: Thanks to the reviewers for their careful review, and we apologize for not describing the purpose and conclusion of this paper carefully. Regarding this issue, we rediscussed and revised it in the introduction and conclusion. Finally, thank the reviewers for their careful reading and review.

  1. To make the manuscript apt for publication, a thorough reorganization must be done. In particular, more references have to be included; the section in which material and methods are described has to be rearranged; original data must be reported (number of samples, characteristics, ...); the results obtained should be properly presented, together with more readable (and error-free) figures. The innovative character (if any) of the paper should also be properly described. In particular, in the first part of the paper the Authors prove the better performance of the random forest (and logistic regression) with respect to other methodologies. But, according to the results reported in Table 2, it appears that the predictive ability of all the methodologies is, in any case, not high enough, and could compromise real applications of pipe replacement planning and risk assessment. Such issues must be discussed in more detail by the Authors in their manuscript.

Response: Thanks to the reviewers for their careful review and helpful suggestions. We have added new recent references, which can be found in the references section. We have reorganized and improved the Materials and methods section in the revised manuscript, providing more detailed raw data (sample size, characteristics, etc.) and a clear method description. The original data will be sent to you together with the revised manuscript. In section 4.2, the methodology is perfected, and in section 4.3, the concept of hyperparameters and the reasons for optimizing the model are perfected. We have presented the results more clearly in the revised manuscript and corrected the figures to improve readability and accuracy. For example, some unclear content and abbreviations can be modified in Table 1. Added the number of original samples, original majority class samples, and original minority samples in lines 305 to 308. Corrected the error of 'interation' in Figure 4. Added explanation of interaction analysis plots in Section 4.5.2, lines 400 to 401. Corrected the x-axis error in Figure 11. Added explanations for NOPF_No_0 and NOPF_No_1 in Figure 12 in Section 4.5.3, lines 443 to 444. Regarding the innovation of the paper, we have revised and supplemented lines 484 to 509 of the conclusion. As for the predictive ability of the model, we can see from Figure 15 that the predictive ability of the model with previous pressure is stronger than that without previous pressure.

Other specific issues:

  1. References should be integrated with more citations related to the subject of pipe replacement planning and pipe burst predictions. Moreover, bibliography is not up to date (actually, citations of any papers from years 2021, 2022 and 2023 are missing).

Response: Thanks to the reviewers for their careful review and helpful suggestions. We acknowledge the paucity of literature on pipeline replacement planning and pipeline rupture prediction in the current reference literature, as well as the lack of literature in recent years. To make the references more complete and time-sensitive, we have searched and cited relevant documents in 2021, 2022, and 2023.

  1. On page 2, line 88, ref. [39] has no meaning.

Response: Thanks to the reviewers for their careful review and pointing out problems. This issue was an oversight by the us, resulting in an incorrect citation, and we have revisited and re-reviewed this section. Thanks again to the reviewers for their valuable suggestions, and express that you will give full consideration to these suggestions in the revised article to ensure that the citations of the article are accurate.

  1. On page 6, Section 4.2, the functioning of the SMOTE algorithm is not clear: the Authors should give a detailed explanation or, at least, a useful reference for understanding it. Moreover, the Authors should also include the details of the original dataset: total number of samples and relative percentage of negative and positive ones.

Response: Thanks to the reviewers for their careful review and helpful suggestions. We acknowledge that the functional description of the SMOTE algorithm may not be clear enough in the current article. In addition, the original data set contains 5376 samples, of which 55 are minority samples and 5321 are majority samples, the ratio of the two is about 100:1. Thanks again to the reviewers for their valuable suggestions, and express that you will give full consideration to these suggestions in the revised article to ensure that the citations of the article are accurate.

  1. Figure 4 on page 7 contains errors: 'interation' should be 'iteration'.

Response: Thanks to the reviewers for their careful review and pointing out problems. We acknowledge the problem with Figure 4. We have modified Figure 4 as follows:

Thanks again to the reviewers for their valuable suggestions, and express that you will give full consideration to these suggestions in the revised article to ensure that the citations of the article are accurate.

  1. On page 11, the Authors have to describe the meaning of the interaction plots. Moreover they must explain the meaning of 'pressure drop': is it a pressure difference or a total head difference between pipe nodes?

Response: Thanks to the reviewers for their careful review and helpful suggestions. We acknowledge that the implications of interaction plots are not adequately described in the current article. Therefore, we modify lines 400 to 401 as follows: the darker the pipeline in the interactive analysis diagram, the greater the probability of failure, and the size of the circle indicate the number of pipelines here. 'pressure drop' explains the pipe pressure at each measuring point.

  1. Figure 11 on page 12 contains errors in the x-axis. The Authors should better explain the meaning and the importance (if any) of the Partial Dependency Picture (PDP), illustrated in Figures 11 and 12.

Response: Thanks to the reviewers for their careful review and helpful suggestions. We acknowledge that there is indeed an error on the x-axis of Figure 11 and will examine Figure 11 carefully and correct the error. For the interpretation of the partial dependence graph, we explained it in lines 429 to 437 in the text.

  1. Comments on the Quality of English Language Extensive English editing is required: many sentences miss the verb or are gramatically incorrect. Others are difficult to understand in their meaning.

Response: Thanks to the reviewers for their careful review and honest feedback. We acknowledge that in the current article, there are indeed some English language quality issues, including missing verbs, grammatical errors, and unintelligible sentences. To address these issues, articles will be carefully edited and polished to ensure improved language quality. Finally, I would like to thank the reviewers again for their valuable suggestions and express that we will fully consider these suggestions in the revised article to ensure that the article conveys the research content more clearly and accurately and improves the quality of the English language.

We also found a native English editor to modify the grammar and sentence structure in the article, which improved the language expression of the article. The attachment is an Editorial Certificate about this article.

Round 2

Reviewer 1 Report

The quality of the manuscript has been improved much. However, it can be further improved in the following aspects.

(1) The data in the case study area should be described in more details, such as the total length of the pipes, the total number of damages, the data collection period, etc. 

(2) The authors are encouraged to revise most of the figures to make them self-evident. Such as in Fig 9, what does average age mean? In Fig 10, what is the y axis? In Fig 11, what is the y axis, and what does the three blocks mean after the "Mat" (same as Fig 13)?  Another example is Fig 8, why use "prediction dis" rather than "prediction distribution"? In my opinion, almost all the figures need to be improved. 

(3) The conclusions need to be further consolidated. What's more, it is just a summarisation of the conducted work, and there is not any quantitative conclusion, which is more important. 

The naration can be more consise.

Author Response

Dear Reviewers:

Thank you for your letter and for the reviewer's comments concerning our manuscript entitled “Application research on risk assessment of municipal pipeline network based on random forest machine learning algorithm” (ID: water-2378527). Those constructive comments are all valuable and very helpful for revising and improving our papers well as the important guiding significance to our research. We have studied the comments carefully and have made corrections which we hope meet with approval. We have added line numbers to the revised manuscript to facilitate the search for specific revised positions. And “Track Changes (second)” versions are also submitted for comparison of changes before and after.

Reviewer #1: The quality of the manuscript has been improved much. However, it can be further improved in the following aspects.

(1) The data in the case study area should be described in more details, such as the total length of the pipes, the total number of damages, the data collection period, etc.

Response: The total length of the pipeline, the diameter of the pipeline, the monitoring period of the pipeline and the age of the pipeline are shown in the line 321 to 326, and the number of pipeline damages is shown in Section 4.5.1 or Figure 6. In addition, we have added lines 408 to 409 of Section 4.5.2, lines 445 to 448 and lines 454 to 457 of Section 4.5.3.

(2) The authors are encouraged to revise most of the figures to make them self-evident. Such as in Fig 9, what does average age mean? In Fig 10, what is the y axis? In Fig 11, what is the y axis, and what does the three blocks mean after the "Mat" (same as Fig 13)?  Another example is Fig 8, why use "prediction dis" rather than "prediction distribution"? In my opinion, almost all the figures need to be improved.

Response: In the line 436 to 438 of the paper, it is mentioned that the PDP chart is only used for trend analysis rather than quantitative analysis, so Figure 11 and Figure 12 have no specific meaning. In other figures, we added "number of pipes" to the x-axis in Figure 3a, and "pipe age" to the x-axis in Figure 3b to represent the number of pipes and the age of pipes respectively; Added "count of damage pipe" and "average target" to represent the number of damaged pipes and the proportion of each pipe age to the total; in Figure 7a, we modified the content of the y-axis and changed it to "diameter" as Pipe diameter; in Figure 8 we changed the y-axis to "prediction distribution"; in Figure 9 we changed it to "pipe age"; in Figure 10 we changed the y-axis to "measuring point flow"; in Figure 11 we changed the title to "mat_ductile iron"; in Figure 13, we changed the title to "PDP interact for "Mat_steel" and "age"" and "number of unique grid points:(Mat_steel;2, age: 10)"; in Figure 14, " 39=len" is changed to "39=length", and "100=Dia" is changed to "100=diameter".

(3) The conclusions need to be further consolidated. What's more, it is just a summarisation of the conducted work, and there is not any quantitative conclusion, which is more important.

Response: We have revised the fifth section, the conclusion, and the main content is found in lines 500 to 516.

Reviewer 3 Report

The Authors have corrected their manuscript according to the suggestions provided

Author Response

Thank you for your recognition of the manuscript. We will continue to work hard to improve the quality of the manuscript!

We have conducted another minor revision this time. Please review the latest submitted manuscript!